



# Flood and drought risk assessment for agricultural areas (Tagus Estuary, Portugal)

Paula Freire[1], Marta Rodrigues[1], André B. Fortunato[1], Alberto Freitas[2]

[1]Laboratório Nacional de Engenharia Civil, Lisboa, 1700-066, Portugal
[2]Direção Geral de Agricultura e Desenvolvimento Rural, Lisboa, 1949-002, Portugal

*Correspondence to*: Paula Freire (pfreire@lnec.pt)

**Abstract.** Estuaries are coastal systems particularly vulnerable to climate change effects and within these systems, agriculture is one of the most potentially affected sectors. This paper proposes a risk assessment approach for helping the decision-making process at a local level, addressing two risks that affect agricultural areas located in estuarine margins: the
unavailability of fresh water for irrigation resulting from the upstream propagation of estuarine brackish water during droughts, and land inundation by high water levels associated with high tides and storm surges. For each risk, quantitative consequence descriptors are presented to support risk level determination and evaluation through a continuous consequence/probability diagram. The approach applicability is discussed through its application to the Lezíria Grande de Vila Franca de Xira, located in the Tagus Estuary (Portugal). Results indicate that the approach is appropriate to support risk
owners in taking actions to mitigate the risk. The flexibility of the approach to be adapted to local conditions and updated through time, and the ease of its application by the risk owner can be pointed out as the main strengths.

## 1 Introduction

Agriculture is one of the economic sectors most vulnerable to climate change effects (Gornall et al., 2010; Burke and Emerick, 2016; Thornton et al., 2018). Extreme weather events, such as floods and droughts, coupled with changing rainfall
patterns, increasing temperatures and rising water demand can reduce crop productivity, as already observed in some southern European countries (Calzadilla et al., 2013; IPCC, 2014; Kovats et al., 2014; European Environment Agency, 2019). In coastal areas, agriculture is experiencing negative impacts mostly associated with the increase of submersion frequency by salt water (IPCC, 2014). Under the influence of both marine and freshwater environments, estuaries are particularly affected by changes in climate, through mean sea level rise, increasing storminess, global warming and
dwindling precipitation (Wong et al., 2014). The development of mitigation and adaptation measures to reduce the impacts of climate change in the agricultural sector is one of the EU Common Agricultural Policy priorities (European Union, 2019). Risk management approaches are increasingly used to help stakeholders in decision-making (Plate, 2002; Ale et al., 2015; Aven, 2016). Risk management aims at anticipating and preventing or mitigating harm that can be avoided, by ensuring that significant risks are identified and reduced through appropriate measures (Simonovic, 2012). The risk management process



should incorporate evidence-based information in supporting the definition of mitigation and adaptation measures (ISO, 2009b). UNISDR (2017) argues that an effective risk management should be based on an understanding of risk from all sources and of the links between hazards and vulnerabilities. Recognizing the complexity of the risk management process, different national and international guidelines have been produced (e.g. AS/NZS, 2004; IRM, 2002). Among them, the ISO 31000 (ISO, 2009) provides generic guidance for the adoption of a consistent process to ensure effective risk management.

This Standard presents a comprehensive framework, which structures the risk management process through five main steps: establishing the context, risk assessment, risk treatment, communication, and monitoring and review. Risk assessment outcomes support the design of risk mitigation measures, their implementation and their effectiveness assessment. While this framework is useful to guide the applications throughout the risk management process, it remains very generic. Hence, its operational implementation needs to be detailed for each specific application. A wide range of approaches for assessing risk

have been developed, including qualitative, semi-quantitative or quantitative techniques (ISO, 2009b; Marhavilas et al., 2011; Sun et al., 2020). Chemweno et al. (2018) discuss the extent of application of several approaches dependent on failure dependencies and on the uncertainty often associated with the lack of reliability data. A comparative review of risk assessment and management methodologies addressing hydro-meteorological natural hazards emphasizes the wide range of approaches followed, as well as their development level and complexity, mostly depending on the location and target

subjects (Cirela et al., 2014). Nevertheless, approaches addressing challenges that climate change will bring to the agricultural areas located in estuarine margins and suitable to support local decision-makers to manage risk remain to be developed. Agricultural estuarine areas without water storage capacity and located in low elevation terrains are particularly vulnerable to changes in water availability for irrigation and inland inundation.

The present study aims at developing a risk assessment approach considering two natural risks that affect agricultural

estuarine lowlands: the scarcity of fresh water for irrigation and the marine submersion. Both phenomena are not new, but they are exacerbated by climate change, through more frequent and intense droughts, increasing storminess and sea level rise. The new approach is applied to an agricultural area (Lezíria Grande Public Irritation Perimeter) located in the Tagus estuary (Portugal). The approach addresses two main challenges: (1) to assess two risks that affect estuarine agricultural areas with different temporal scales of consequences (the scarcity of fresh water for irrigation and estuarine inundation of

agricultural terrains); (2) to consider hazard uncertainty in the risk evaluation. The final goal is to contribute with a tool that can support the decision-making process at a local level in order to manage risk.

The paper is structured in five sections besides this introduction. Section 2 presents the risk assessment approach proposed and Sect. 3 characterizes the study area where the approach is applied. Results of the approach application are described in Sect. 4 (Risk context) and Sect. 5 (Risk assessment). Results are discussed and the main conclusions summarized in Sect. 6.





## 2 Risk assessment approach

A risk assessment approach is developed to address two natural hazards that often affect agricultural areas located near estuaries, particularly those dependent on surface water for crop irrigation and presenting low topography. These hazards are: i) water salinity increase due to droughts, and ii) estuarine high water levels that can promote inland inundation. In order to support stakeholders and decision-makers in the definition of mitigation and adaptation strategies, the approach should be easy to perceive by the stakeholders and suitable to be updated according to local conditions. The approach is summarized in Fig. 1 and is based on the generic risk management framework of ISO 3100 (ISO, 2009). The definitions used herein are adapted from ISO (2009a).

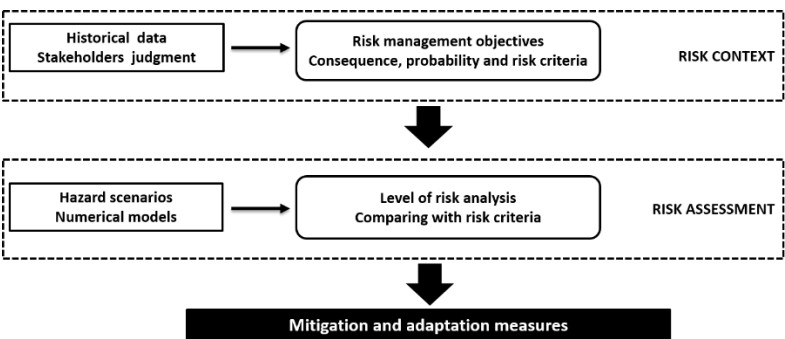

**Figure 1: Risk assessment approach followed in the present study.**

The risk assessment has to be preceded by the establishment of the risk context, which defines the risk management objectives, the consequence descriptors and the criteria to grade consequences, likelihood and risk (ISO, 2009b). The risk context depends on the site-specific characteristics and must be supported by historical information, and stakeholders and the risk owner judgement. The risk owner is the person or entity responsible for the risk management (ISO, 2009a).

As discussed above, several approaches are available to operationalize risk analysis and evaluation. Risk matrices, combining qualitative or semi-quantitative information on consequence and probability, are used in several risk management standards and guidelines to rank and prioritize risk (ISO, 2009b). Despite several disadvantages pointed out in the literature (e.g. Cox, 2008; Duijm, 2015), risk matrices are widely used in risk acceptance discussion and risk communication to broader audiences, supporting decision-making, as they present complex concepts in a simple way (Woodruff, 2005; Ale et al., 2015). As an adequate tool to deal with risk level uncertainty, in both consequence and expected frequency, a continuous consequence/probability diagram is chosen in the present study as the suitable technique to assess risk.

The consequence is defined as an event outcome that affects the risk management objectives (ISO, 2009a). The proposed approach defines quantitative consequence descriptors of the two hazards through indicators of the potential economic impact for the risk owner.





For the water salinity increase during droughts, the consequence descriptor was defined as the water unavailability for irrigation during the most critical period for the crops to be watered. The water unavailability for irrigation ($Wu$) is given by Eq. (1):

$$Wu = 1 - \left[\frac{volume\ of\ water\ available\ with\ salinity < 1psu}{volume\ of\ water\ needed}\right]$$
(1)

Concerning estuarine high water level, several elements are exposed to hazards such as the land, people or infrastructures including dykes that prevent lowland inundation during high spring tides. When dykes are present, inundation normally occurs when the water level is above the dyke crest or when the dyke is breached. This exposed element can provide a direct quantification of the hazard economic impact for the risk owner. Thus, the high water level consequence is based on the dyke overflowing and the chosen descriptor is the relative cost of dyke damage repair, considering that the risk owner is the

organization responsible for repairing the dykes. The relative cost of dyke damage ($RCDD$) given by Eq. (2):

$$RCDD = \frac{length\ of\ the\ affected\ dyke\ x\ repair\ cost\ per\ unit\ length}{risk\ owner\ annual\ income}$$
(2)

Criteria to grade the consequence severity should rely on past events information from the area where this approach is applied, with the stakeholder's involvement. The same must be followed when selecting classes of likelihood, defined as the chance of something happening and can be presented as a probability of an event.

In the present approach, the definition of risk levels considers the ISO (2009a) criteria and the tolerable risk concept that is normally used to assist decision-makers (Marszal, 2001). Tolerable risk is defined by ICOLD (International Commission on Large Dams) in 2002 as "a risk within a range that society can live with so as to secure certain net benefits". It is a range of risk that cannot be neglected or ignored and should rather be kept under surveillance and reduced if possible (Bowles, 2003). Below tolerable risk, the risk is acceptable, i.e. risk is considered insignificant or adequately controlled, and above risk is

unacceptable (HSE, 2001). For the hazards considered, risk is divided in three levels in the consequence/probability diagram corresponding to different bands: a) high risk (red band), where the level of risk is considered intolerable and risk treatment is essential whatever its cost; b) medium risk (yellow band), where the risk is considered tolerable; c) low risk (green band), where the level of risk is considered negligible, so no risk treatment measures are needed. Risk tolerance limits depend on the study area characteristics and should be defined based on information from past events and risk owner judgement.

After establishing the risk context for the area where the approach is applied, hazard scenarios based on historical data and stakeholder's information have to be defined to support risk assessment. Consequence descriptors are evaluated for the defined scenarios and risk levels are determined, compared and evaluated against risk criteria and tolerance limits previously defined. Results provide scientifically-supported information to help stakeholders and risk owners to discuss the acceptability of the risk magnitude. The consequence descriptors can be evaluated through the analysis of model results, and

historical and monitoring data.



## 3 Study area

### 3.1 The Tagus estuary

The Tagus estuary, located at the mouth of the Tagus River basin (Fig. 2), is framed by the largest metropolitan area of Portugal, hosting along its margins 1.6 million inhabitants (Tavares et al., 2015). With a surface area of about 32,000 ha, the
estuary presents a marked contrast of occupation between both margins: extensive artificial areas are present along the northern margin and agricultural and semi-natural areas including a Natural Reserve (one of the most important sanctuaries for birds in Europe with about 14,000 ha) in the eastern area. Agriculture is the most relevant economic activity in the Tagus estuary upper region, in particular irrigated agriculture. Two different types of water resources management are present: the collective management existing in the irrigation perimeters of state / public initiative, either through distribution from
reservoirs (Vale do Sorraia) or through direct extraction from the Tagus River (Lezíria Grande de Vila Franca de Xira); and individual management carried out by farmers outside these perimeters.

The main source of freshwater discharging into the estuary is the Tagus River, with an average, maximum and minimum annual flows of 336 $m^3s^{-1}$, 828 $m^3s^{-1}$ and 102 $m^3s^{-1}$, respectively (APA, 2012). The Sorraia and the Trancão rivers also contribute to the freshwater inflow into the estuary. The Tagus is the longest river of the Iberian Peninsula with a watershed
of 80,100 $km^2$ distributed between Portugal (30%) and Spain (70%). The hydrological regime is highly modified by several reservoirs constructed since the 1950's in both countries, along the Tagus River and its tributaries. Although a convention was signed between the two countries in 2001 to agree on annual water releases in the Tagus River at the international border, particularly during droughts these releases are irregular and difficult to account for (Henriques, 2018). Therefore, the water availability downstream strongly depends on the water resources management practices in the basin.

The hydrodynamics of the Tagus estuary is primarily driven by tides. The tidal range varies between 0.55 m and 3.86 m at the coast (Guerreiro et al., 2015), and increases inside the estuary due to resonance (Fortunato et al., 1999). During extreme conditions, other forcings may also be important. High river flows can increase water levels in the riverine stretch of the estuary (Vargas et al., 2008) and stratify the water column (Rodrigues and Fortunato, 2017). During storms, wind, atmospheric pressure and surface waves can also increase the water levels significantly (Fortunato et al., 2017).

The upper part of the estuary is affected by natural hazards with different meteorological and oceanographic origins, often with relevant socio-economic impacts. Droughts can result from extremely dry periods aggravated by the impact of the water management practices in the Tagus river basin. These water scarcity events significantly reduce the river flow reaching the estuary and consequently increase the saltwater intrusion, as observed in 2005 and 2012 (Rodrigues et al., 2019). The vulnerability of the water for human consumption, in terms of quantity and quality, was assessed for the EPAL (Public
Water Supply Company) water intake located in the Tagus estuary upper sector (Valada do Tejo) for different climatic scenarios (Rodrigues et al., 2012). Both the results of that study and those of Rodrigues et al. (2019) suggest that only very low river flows would lead to a significant increase of the salinity in the area. Historical data show that the Tagus estuarine margins are also vulnerable to floods from two different origins that can widely affect the agricultural lands due their low





elevation (Freire et al., 2016; Rilo et al., 2017): extreme water discharges in the Tagus and Sorraia rivers (riverine flood),

and strong winds and low atmospheric pressure conditions combined with high spring tides (Fortunato et al., 2017).

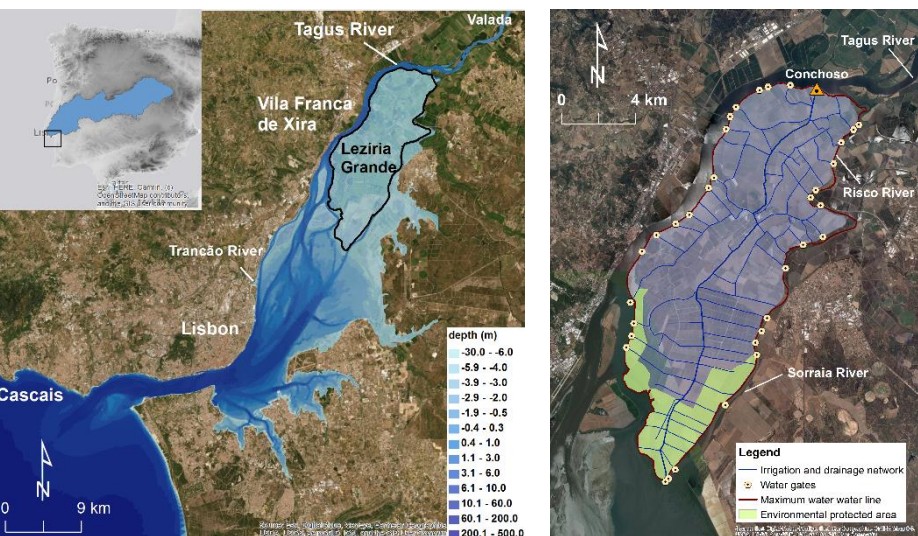

**Figure 2: Location and place names of the study area: bathymetry of the Tagus Estuary, and location of the Tagus watershed in the Iberian Peninsula (inset) (left panel); irrigation network in the Lezíria Grande Public Irrigation Perimeter (right panel) (ESRI**
**Basemaps).**

### 3.2 Lezíria Grande Public Irritation Perimeter

The Lezíria Grande de Vila Franca de Xira Public Irrigation Perimeter (Lezíria Grande) is an important economic agricultural area, located on the Tagus estuarine eastern margin, about 40 km from the estuarine mouth (Fig. 2). This very productive area with 13,420 ha of alluvial soils of both fluvial and estuarine origins belongs to the Metropolitan Area of

Lisbon and is part of the municipalities of Vila Franca de Xira and Azambuja. The Lezíria Grande occupies low elevation terrains, between mean sea level (MSL) and 2 m above MSL, reclaimed from the estuarine bed and protected from flooding by a 62 km long system of dykes, along the margins of the Tagus, Sorraia and Risco rivers. The dykes are made of soil covered by vegetation and in some places their outer flanks are protected with riprap. Available topographic data indicate that the dykes crest reach heights between 2.4 and 7.2 m above MSL. The southern area of the Lezíria is part of the Tagus

Estuary Natural Reserve. The Lezíria Grande has a relevant impact on the local and regional economies, with an annual investment in crops of about 40 million euros and involving about 6,000 direct jobs (https://www.publico.pt/2005/08/26/jornal/fecho-do-rio-sorraia-salva-culturas-da-leziria-grande-de-vila-franca-36092, accessed on May 2020) during the Spring-Summer agricultural season and some additional indirect jobs related to services and equipment. The main crops are rice, cultivated in the downstream area due to its higher tolerance to salty water, tomato

and corn, which jointly represented 91% of the cultivated area in 2017.





The Lezíria Grande presents a complex irrigation and drainage system network of channels 720 km long that are connected to the adjacent rivers (Tagus, Sorraia and Risco) by water intakes and drainage gates. The main water intake that supplies the freshwater for the farmland irrigation is located in the Tagus River, at Conchoso, and includes a pumping station (Fig. 2). The total irrigated area is about 10,000 ha, 60 % of which are irrigated by surface irrigation and 40 % under pressure.

As the Conchoso water intake is located close to the upstream limit of the salinity propagation, the availability of water with quality for irrigation strongly depends on the freshwater input from the Tagus River into the estuary. Because the effect of droughts in the freshwater input usually starts in July, the critical month for irrigation, crops can be lost, with relevant economical losses. During the most recent severe droughts, in 2005 and 2012, several emergency measures were undertaken in the Lezíria Grande to minimize the negative impacts, such as the water supply exclusively from the Risco river water

intake and the construction of a temporary weir at the Sorraia river. The installation of the pumping system at the Conchoso water intake, allowing the extraction of water from the Tagus River during low tide, and the construction of a removable weir in the Risco River are recent improvements to increase the resilience to droughts.

Due to its low elevation terrains, the Lezíria Grande is vulnerable to flooding episodes of both riverine and estuarine origins. High water discharges of the Tagus and Sorraia rivers can promote dyke breaching and extensive agricultural lands

inundation as occurred in February 1979 (Rebelo et al., 2018). During this event, the Lezíria Grande dyke was ruptured both in the north and south sides, originating either displaced or evacuated people and relevant economic losses. About 2,000 people were reported to have been affected due to dyke failures in the surrounding area (Loureiro, 2009).

Estuarine high water levels caused by spring tides and severe storm surges can also overflow and damage the Lezíria Grande dykes, promoting extensive inland inundation. The most dramatic estuarine flood event that affected this area, destroying all

the channels and dykes, occurred on February 15, 1941 (Madaleno, 2006). This event affected several locations along the Portuguese coast with devastating human and physical impacts (Muir-Wood, 2011; Freitas and Dias, 2013) and is considered the major calamity that affected the Iberian Peninsula in the last 200 years. Besides the severe damages in infrastructures, the impacts in the upper estuary include human losses and drowned cattle (Muir-Wood, 2011). The most recent estuarine flood event that affected the Lezíria Grande occurred on February 27, 2010 resulting from the passage of the storm Xynthia in the

Portuguese territory (André et al., 2013; Fortunato et al., 2017). The dykes in the southern area of the Lezíria Grande were overflown and damaged and the farmland flooded. As the event occurred out of the active farm season, witnesses report that only up to 5 families and some cattle had to be evacuated. After the event the dykes were repaired and elevated in some places.

## 4 Risk context

### 4.1 Risk management objectives

The Associação de Beneficiários da Lezíria Grande de Vila Franca de Xira (ABLGVFX), a collective organization responsible for the management of the public infrastructures, under the supervision of the General Directorate of Agriculture



and Rural Development (DGADR), acted as risk owner as they are the most representative stakeholder. The overall risk management objective of the Lezíria Grande is the management and exploitation of a public irrigation infrastructure during extreme weather conditions. These conditions can be aggravated by climate change effects, namely more extreme droughts and floods. For the present application and considering the saltwater intrusion hazard, the risk management objective is to ensure water with good quality at the Conchoso intake, i.e. water with salinity below 1 psu, during the agricultural irrigation campaigns. Despite the natural conditions that contribute to the droughts, the water resources management practices at regional and local levels affect this objective. At the regional level, the volumes discharged from Spain during exceptional meteorological conditions and the EDP hydropower production regime are the main conditioning factors for the water availability downstream. The adaptive capacity of the farmers, such as improving the adequacy and efficiency of the irrigation practices, changing the type of cultures, and increasing emergency planning and response capability are examples of local water resources management factors. Concerning the high water level hazard, during estuarine floods, the specific risk management objective considered is to avoid the dykes overflow and damage, preventing inland inundation. Due to the Lezíria Grande low topography, the dyke integrity is crucial to protect the farmland, support facilities and infrastructures from being inundated and damaged, not only during extreme events but also daily during high tide. This objective can be reached by flood adaptation measures, including raising the dykes height, as decided after the 2010 flood event, increasing the area of salt tolerant crops, and increasing emergency planning and response capability.

## 4.2 Consequence and likelihood criteria

For the water salinity at Conchoso during droughts, which results from the upstream saltwater propagation, the consequence is evaluated using the consequence descriptor presented in Eq. (1), considering the water unavailability for irrigation during the month of July as it is the most critical for irrigation. Water unavailability is computed on a weekly basis and the minimum weekly water need is considered as $1,029 \times 10^3$ m$^3$, which corresponds to the worst case scenario based on historical needs (Aqualogus/Campo d'Água, 2016). The usable volume of water is estimated by multiplying the time during which water with salinity below 1 psu is available at the Conchoso intake per week by the maximum pumping capacity at the Conchoso station. The Conchoso pumping rate capacity considered was 4.5 m$^3$s$^{-1}$, which corresponds to the pumping rate capacity with low water level (Aqualogus/Campo d'Água, 2016a). This criterion is justified by the absence of reservoirs in the Lezíria and it is assumed that the water is used as soon as it is abstracted. The severity grade criteria of this consequence were defined based on the past occurrences and their consequences (Table 1).

During the most recent droughts, the consequences were more severe in 2005 than in 2012 as less water was available. In 2005, fresh water was unavailable at the Conchoso water intake from mid-July onwards. The water was therefore supplied to the Lezíria exclusively from the Risco River water intake and the consequences were very severe with significant losses of crops (Rodrigues et al., 2016). Thus, severity is considered low when less than 1 % of the water needed is unavailable for irrigation, leading to negligible losses of crops. The severity is considered medium when 1 %-25 % of water is unavailable,



while high severity corresponds to 25 %–50 % of water unavailable for irrigation. Very high severity corresponds to over 50 % of the water unavailable, which can lead to very significant losses of crops and, consequently, economical losses.

For the high water level, the consequence is evaluated based on the descriptor presented in Eq. (2), where the repair cost per unit length is estimated based on the values of the dyke repair cost and affected length during the event of February 2010 described in Sect. 2. The ABLGVFX annual income is averaged over 2014-2018 to reduce the sensitivity to inter-annual variations. All values were provided by the ABLGVFX and updated to 2019. The criterion to grade the consequences affecting the risk management objective is to avoid the dykes being overflown and damaged. This criterion was defined based on the impact of the dyke repair cost on the risk owner annual income (Table 1). Severity is considered low when the dyke repair cost is less than 1 % of the annual income, which corresponds to twice the dyke annual maintenance cost. Considering that the impact of the February 2010 storm event, which was about 4 % of the annual income, has a medium-low severity, the upper limit of this class is defined as 10 %. Very high severity consequence is considered when the dyke repair cost exceeds 30 % of the annual income. The likelihood criteria for both hazards are presented in Table 2.

**Table 1: Grade of consequences for water salinity and water level.**

| Severity | Unavailable water for irrigation | Relative cost of dyke damage |
|---|---|---|
| Low - L | $\leq 0.05$ | $\leq 0.01$ |
| Medium - M | 0.05–0.25 | 0.01–0.1 |
| High - H | 0.25–0.5 | 0.1–0.3 |
| Very high - VH | > 0.5 | > 0.3 |

**Table 2: Criteria for the likelihood.**

| Likelihood | Probability of occurrence/year | Return period (year) |
|---|---|---|
| Very low - VL | 0–0.01 | > 100 |
| Low - L | 0.01–0.1 | 10–100 |
| Medium low - ML | 0.1–0.2 | 5–10 |
| Medium - M | 0.2–0.5 | 2–5 |
| High - H | 0.5–1 | 1–2 |

### 4.3 Risk criteria

For the water salinity, risk tolerance limits are defined based on the water availability and the possibility to fulfil the needs from alternative water sources (Table 3). Risk is considered low when the water available at the Conchoso water intake is sufficient to meet the irrigation needs. Thus, the criterion followed to define the upper limit of the low risk is: the water unavailable for irrigation is less than 1 % for high likelihood events (i.e., events with a return period RP=1 year). Medium





risk level, which corresponds to the tolerable risk, corresponds to events during which the water available at Conchoso cannot meet the irrigation demands, but the Risco River can be used as an alternative source to fulfil the needs. The upper and lower limits of the tolerable risk band were defined based on estimates of the minimum and maximum volumes of water available in the Risco River. The minimum volume of water available in the Risco River was defined based on Rodrigues et al. (2019), which estimates that the volume of water available in the Risco River ranges from 1–4x$10^6$ m$^3$. Considering that the Risco River should provide an alternative water source during one month, the minimum weekly volume available in the Risco River is 0.25x$10^6$ m$^3$, which corresponds to 24% of the total water needs for irrigation per week. The maximum water volume available is determined by the water abstraction capacity. The average abstraction capacity was taken as 0.97 m$^3$ s$^{-1}$ (Aqualogus/Campo d'Água, 2016), which corresponds to a weekly volume of 584,558 m$^3$ (57% of the irrigation needs). Thus, the risk is considered tolerable if the water unavailable in Conchoso is less than 24% (for high likelihood events, i.e., RP=1 year) or 57% (for low likelihood events, i.e., RP=100 years). The risk is considered high when the water available from both Conchoso and the Risco river is insufficient to fulfil the irrigation needs and risk treatment is required.

Table 3: Risk level criteria for water salinity. Colour code refers to Fig. 4.

| Risk level / Colour code | Risk criteria |
| --- | --- |
| Low / Green | Water available in Conchoso is sufficient for the irrigation needs |
| Medium / Yellow | Water available in Conchoso is not sufficient for the irrigation needs and the water from the Risco river is used as an alternative source |
| High / Red | Water available in Conchoso and Risco river is not sufficient for the irrigation needs |

For the high water level, risk tolerance limits are defined based on the potential impact of the dyke damage cost on the risk owner annual profit, measured by the ratio between the repair cost and the risk owner annual income. Risk is considered low when the damage repair cost is negligible relative to the annual income. Thus, the upper limit of the low risk band is defined as: the cost of the dyke repair does not exceed the annual dyke maintenance cost, that represents about 0.5 % of the annual income, for high likelihood events with RP=1 year, and the double for events with low/medium low likelihood (RP=10 years) (Table 4). Medium risk level corresponds to the tolerable risk, i.e., the impact of the dyke damage repair cost on the annual income is tolerable for the risk owner. The lower limit of the tolerable risk band corresponds to the upper limit of the low risk. The upper limit of the tolerable risk is defined by considering that the dyke repair cost should not endanger the financial viability of the risk owner. The impact of the February 2010 event, already presented, is used to help defining this limit: for high likelihood events (RP=1 year), the risk is considered tolerable if the repair cost does not exceed 4 % of the annual income, and 4 times more in case of events with low/very low likelihood (RP=100 years) (Table 4). Above the tolerable risk upper limit, risk is considered high and unacceptable, and in this case risk treatment is required whatever its cost to reduce the risk level.



**Table 4: Risk level criteria for high water level. Colour code refers to Fig. 6.**

| Risk level / Colour code | Upper limit criteria | | Criteria |
|---|---|---|---|
| | Likelihood (RP in years) | Consequence (dyke repair cost/risk owner annual income) | |
| Low / Green | RP=1 | 0.005 | Dyke repair cost impact on risk owner annual income is negligible |
| | RP=10 | 0.01 | |
| Medium / Yellow | RP=1 | 0.04 | Dyke repair cost impact on risk owner annual income is tolerable |
| | RP=100 | 0.16 | |
| High / Red | | | Dyke repair cost impact on risk owner annual income is negligible is unacceptable |

## 5 Risk assessment

### 5.1 Water salinity

As stated above, the salinity at the Conchoso intake depends mostly on the Tagus River discharge and on the water
management practices in the Tagus watershed. The size and strong artificialization of the watershed, shared between two
countries, make the hydrologic modelling a complex and time-consuming task far beyond the scope of this work. The
capacity of flow regularization in the Spanish part of the basin reduces the average flow at the Spanish-Portuguese border by
27% (Aus Der Beek et al., 2016). Therefore, hazard scenarios were constructed based on available data and on past event
information. July is the critical month for crop irrigation and the upper salinity limit for irrigation is 1 psu. Considering these
conditions, and to provide a wide range of events for the risk assessment, five scenarios of Tagus river discharge were
established (Table 5). A scenario combining the worst recent drought (SD2) with the possible sea level rise of 0.5 m was also
considered. This value is representative of the prediction for the end of the 21st century considering the Representative
Concentration Pathway (RCP) scenarios RCP2.6 and RCP8.5 (Rodrigues et al., 2019). For all scenarios the likelihood was
estimated based on relevant historical events and on probability estimates, and follows the criteria already presented in
Table 2. Further discussions about the scenarios can be found in Rodrigues et al. (2019). The quantitative consequence
descriptor defined previously is assessed for the different scenarios through numerical modelling. Numerical models
implemented and validated for the study area are described in Appendix 1.

Figure 3 presents the different scenarios projected in the consequence/probability diagram for the water unavailability; the
horizontal and vertical bars represent the expected uncertainty for consequence and likelihood, respectively. Consequence is
low for all the scenarios in the first week, since the water available fulfils all the needs for irrigation. As time progresses the





consequence increases for all the scenarios with exception of scenario SD1 (climatological, mean river flow of 136 m$^3$s$^{-1}$), in which water is always available for irrigation. The consequences are also more severe when the river flow is lower, as expected, although very low river flow scenarios (SD4, SD5) have low likelihoods. The estimated consequences for the scenarios agree with the observed occurrences during recent droughts (2005, 2012), as described by the risk owner. During
July and August of both 2012 and 2005, droughts represented by scenarios SD2 and SD3 respectively, salinity reached concentrations at the Conchoso water intake that were inadequate for irrigation. In 2012, in particular, water with salinity of about 1.1–1.2 was used for irrigation, which reduced the production. However, the adverse impacts of the 2005 drought were more severe for the farmers in the Lezíria, since the drought itself was more severe and the ABLGVFX had fewer resources and was less prepared to deal with these events. More severe consequences are also estimated for scenario SD3
comparatively to scenario SD2 (Fig. 3). Since the consequence of all the scenarios is estimated based on numerical simulations there is an associated uncertainty. To estimate the uncertainty of the consequence, the maximum difference between the data and the model results at the peak salinity (2 psu) was used and the estimations described previously were performed considering the water salinity <3 psu. Results suggest that its influence on the consequence severity is higher for low river flow scenarios, and in some cases, consequences can range from "Very high" to "Low". However, it should be
noted that the criteria used to define the uncertainty corresponds to the maximum peak difference, which explains the larger variability in the consequence.

**Table 5: Scenarios for water salinity and corresponding likelihood considering the Tagus river flow (*Q*) and sea level rise (SLR) conditions.**

| Scenario | $Q$ (m$^3$/s) SLR (m) | Description | Likelihood |
|---|---|---|---|
| SD1 climatological scenario | 132 0 | Mean daily discharge at the Almourol station (http://snirh.pt) between 1990 and 2017 during the month of July | Medium |
| SD2 recent drought | 44 0 | River flow representative of one of the recent droughts that occurred in 2012, estimated based on data measured at Almourol (https://snirh.ambiente.pt). | Medium low |
| SD3 worst recent drought | 22 0 | River flow representative of one of the worst recent droughts using data from Matrena and Tramagal stations (https://snirh.ambiente.pt) during July 2005. | Low |
| SD4 minimum river flow | 16.5 0 | Minimum mean weekly river flow that must be guaranteed between July 1st and September 30th near the upstream boundary of the Tagus estuary (Muge) by the revised Spanish-Portuguese Albufeira Convention and Additional Protocol (Portuguese Parliament Resolution n. 62/2008, November 14) | Very low |
| SD5 worst case scenario | 8 0 | Minimum river flow that guarantees the operation of the main thermoelectric power plant in the Tagus River (Pego power plant) | Very low |
| SD6 sea level rise | 22 0.5 | Combination of the worst recent drought with a sea level rise of 0.5 m | Low |



Regarding the risk diagram, results indicate that for all the scenarios with exception of the climatological scenario (SD1) the risk is intolerable in the last week (Fig. 4), showing that the risk increases with the duration of the droughts. When the low river flows occur for several consecutive weeks, even using the Risco River as an alternative source of water for irrigation is not sufficient to meet the water needs for irrigation. Thus, for events similar to these scenarios, risk treatment is mandatory

to reduce risk level and may include the use of alternative water sources, the selection of alternative crops, the reduction of the irrigated area and/or investments regarding water storage.

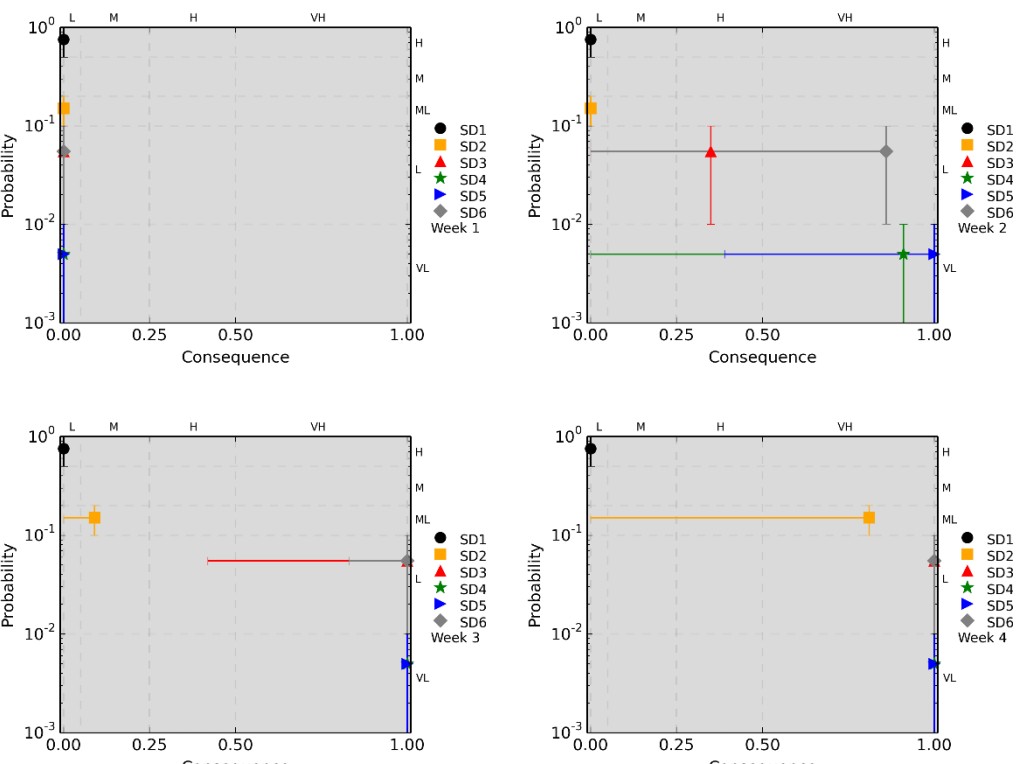

**Figure 3: Consequence/probability diagrams for water unavailability for irrigation.**





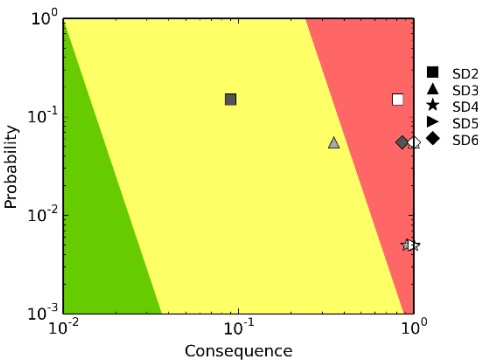

**Figure 4: Risk for water unavailability. Colours of the symbols represent the weeks (darker to lighter means week 1 to week 4).**

## 5.2 High water level

Estuarine high water levels are forced by spring tides and severe storm surges, which are associated with very low atmospheric pressure conditions. Based on the past extreme events of 1941 and 2010, described in Sect. 3.2, that caused overtopping of the Lezíria Grande dykes and inundation of agricultural lands, four scenarios of extreme water levels were defined (Table 6). Extreme water level conditions of the scenarios result from the oceanographic and meteorological conditions of the events. The same sea level rise scenario for the end of the century used for the salinity was considered here (SF4), combined with the storm surge and tide conditions of the 1941 cyclone (SF2). The scenarios were assessed through numerical models implemented and validated for the study area and that are described in Appendix 1. The model estimates of the extent of dyke overflown entails uncertainties associated with several error sources. The tidal levels predicted by the model have errors of the order of 15 cm in the upper estuary, while errors associated with the storm surge can reach about 10 cm (Fortunato et al., 2017). Topographic errors, in particular, in the dykes' crest height, were taken as 10 cm. Taking the overall error as the square root of the sum of the squares of the individual errors leads to a vertical uncertainty of 20 cm. To determine the uncertainty in the estimate of the overflown dyke length, we considered that a difference of 50 cm in water level between two simulations (scenarios S2 and S4 described below) leads to a discrepancy of 130 % in the overflown extent of the dyke. Assuming a linear relationship between the horizontal and vertical dimensions, the uncertainty in the estimate of the length of the dyke overflown is 50 %.



**Table 6. Scenarios for maximum water levels considering different storm surge, tide and sea level rise (SLR) conditions. $W_m$ is the maximum water level at Cascais tide gauge and $Q$ is the Tagus river flow.**

| Scenario | $W_m$ (m, above CD) | $Q$ (m³/s) SLR (m) | Description | Likelihood |
|---|---|---|---|---|
| SF1 storm Xynthia 2010 | 2.24 | 3917 0 | Storm surge and tide conditions observed during the Xynthia storm | Medium low |
| SF2 1941 cyclone | 2.34 | 4517 0 | Storm surge and tide conditions observed during 1941 cyclone | Low |
| SF3 1941 cyclone and spring tide | 2.54 | 4517 0 | Storm surge conditions observed during 1941 cyclone and considering an equinoxial spring tide | Very low |
| SF4 1941 cyclone and sea level rise | 2.84 | 4517 0.5 | Combination of the storm surge and tide conditions of 1941 cyclone with a possible sea level rise for the end of the century | Low |


For all high water level scenarios, the area where the dyke is potentially affected is located in the southern half of the Lezíria (Fig. 5). In scenario S1, about 1 km of dyke near the Lombo do Tejo island is affected. In the scenario SF2, the dyke is affected in the same zone but the length doubles. When a spring tide is considered (scenario SF3) the length of the affected dyke increases up to 4 km, extending the affected area to north of the Alhandra island and to the southern extreme of the

Lezíria. The length of the potentially affected dyke increases to 8 km if sea level rise is considered (SF4).

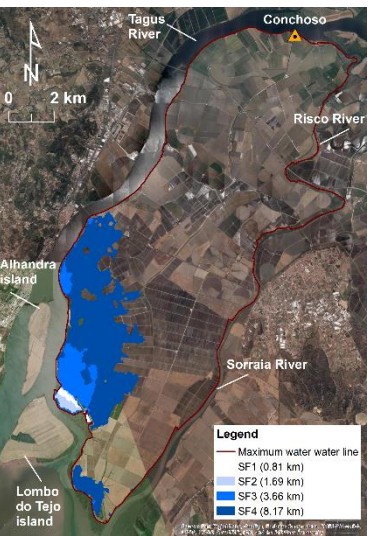

**Figure 5: The Lezíria inundation areas for the hazard scenarios showing the locations where the dyke is potentially affected, and affected dyke length (ESRI Basemap).**






Figure 6 (a) presents the different scenarios projected in the consequence/probability diagram for the relative cost of dyke damage (*RCDD*). Again, the expected uncertainty for both consequence and likelihood is represented by horizontal and vertical bars. The consequence severity of the scenarios with low (SF1) and medium severity (SF2) is consistent with the known impacts of the 1941 and 2010 events, which were much higher in 1941, as described in Sect. 3.2. Medium severity,

corresponding to a dyke repair cost of 1 to 10 % of the ABLGVFX annual income, can be reached for low likelihood scenarios with RP between 10 and 100 years. The consequence severity is "high" (repair cost is up to 30% of the ABLGVFX annual income) for the very low likelihood scenario (scenario 3, RP>100 years). In this case, besides the 1941 storm surge conditions, an extreme tidal range is considered (equinoxial spring tide). Very high consequence severity, expressed by the dyke repair cost over 30 % of the ABLGVFX, is reached if sea level rise is considered (SF4). Limitations of the model can

underpredict the severity level. The model was run with a fixed geometry, i.e., the bathymetry and topography were assumed to remain unchanged during the simulations not considering dykes' erodibility. In reality, events of this type can erode and breach the dykes at several locations, as actually occurred in February 2010, increasing the potential dyke length affected. Because the 1941 scenario is more energetic (in terms of wind speed, water currents and waves), the breaching should also be more severe. Hence, the length affected during this scenario is probably more underpredicted by the model than for the

2010 scenario. None of the scenarios considered has an associated low risk (Fig. 6 (b)), i.e., the dyke is not overflown. This is explained as this risk is not associated with average oceanographic and meteorological conditions. In all scenarios without sea level rise, the risk conditions are moderate (tolerable level), indicating that risk has to be monitored regularly to decide if adaptation measures have to be taken to reduce the risk level. However, as sea level rises, risk will become unacceptable. Hence, risk treatment will be required in the future to bring the risk down to an acceptable level.


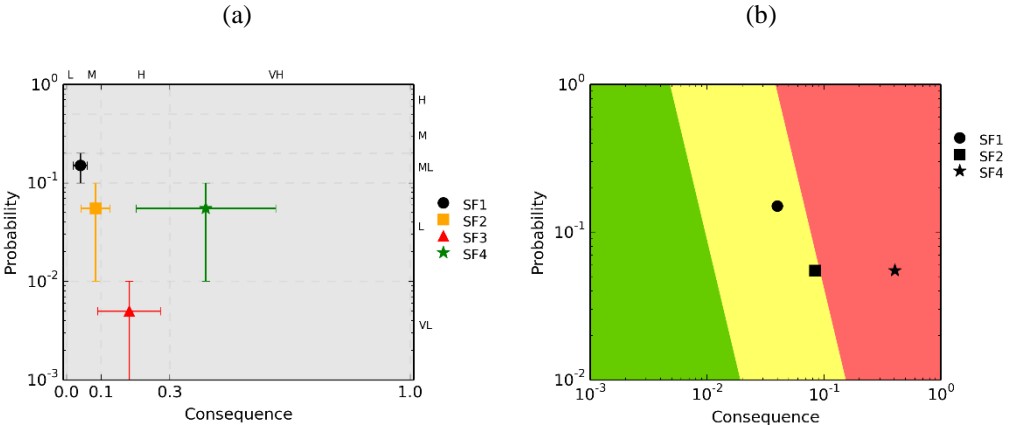

**Figure 6: Consequence/probability diagram for relative cost of dyke damage (a). Horizontal and vertical bars indicate the**
**uncertainty of the hazard in both consequences and likelihood, respectively. Risk diagram for water level relative cost of dyke damage (b).**



## 6 Discussion and conclusions

The risk assessment approach presented in this study intended to integrate the hazard dimensions that most affect agricultural areas located in estuarine margins. Highly dependent on water resources, agriculture is one of the economic sectors most
vulnerable to climate change effects (Aleksandrova et al., 2016). Its vulnerability increases when agricultural areas are located in estuaries where changes in hydrological regimes and sea level rise can impact both salt water landward intrusion and low-lying areas inundation (Kimmerer and Weaver, 2013). The main challenge of the approach developed herein was to find suitable consequence descriptors of the two hazards that incorporate scientific-based data but can easily be applicable by the risk owner and be updated in time. For this, the difference in elements at risk, coverage and temporal scale of impacts for
the two hazards were considered in the definition of consequence descriptors for risk assessment.

For saltwater landward intrusion due to droughts, the water resources availability are the element at risk. The scarcity of suitable water for irrigation has an economic impact for the risk owner, mainly due to crop losses resulting from lack of water or/and salinization of land if salty water is used. Enabling risk owners with tools that anticipate the expected water availability is thus essential to support decision making both before and during the agricultural campaign. An adaptive risk
assessment in a temporal scale of weeks during the most critical period for crop irrigation is suitable for helping manage water unavailability.

For estuarine high water levels associated with tides and storm surges, the elements at risk are mainly the agricultural land itself, and infrastructures such as dykes, support facilities, roads and access infrastructures. Damages in those assets and ultimately the loss of agricultural land due to inundation have a direct economic impact for the risk owner. In agricultural
lands located in low-lying estuarine areas, dykes or other protection structures are normally present to prevent frequent land inundation during high spring tides. Using the relative cost of dyke damage as a consequence descriptor has the advantage to provide a direct quantification of the hazard economic impact for the risk owner, and to be easily estimated through information that the risk owner normally has access to.

Due to the uncertainty of the factors that control both risks, a continuous consequence/probability diagram was found to be
the most adequate technique for risk level estimation and evaluation, as it integrates the uncertainty in the process. In addition, this tool is suitable for communicating the risk in a simple way to the risk owner. The applicability of the developed approach was explored through the application in the Lezíria Grande agricultural area, known to be affected by those two hazards. Results show that concerning fresh water scarcity, the risk increases with the duration of the droughts and when low river flows occur for several consecutive weeks, even using the Risco River as an alternative source of water for
irrigation is not sufficient to meet the water needs. The total dependence of irrigation on the Tagus and Sorraia fluvial discharges, with other users upstream, suggests that previous knowledge of the water availability reserved in Spain and Portugal and the consumption expected for the different sectors upstream is essential in assessing the risk of fresh water unavailability for irrigation. Real-time knowledge of the upstream discharges, existing consumptions and possible runoff


from the rice crops, particularly those located along Sorraia River, will definitely contribute to decision-making regarding
the best periods for estuarine water intake.

Considering the estuarine inundation, the results presented above show that presently the risk in Lezíria Grande is moderate. The hazard can be significant, but only for very extreme events with a high return period. However, sea level rise will increase the risk. Hence, the risk owner should consider risk reduction measures, as they will become necessary in the future. Furthermore, the sea level rise considered herein was based on the 5[th] IPCC assessment report (IPCC, 2014). Since that
report was published, several studies indicate that sea level may rise faster than anticipated (Shepherd et al., 2012; Khan et al., 2014; Scambos and Shuman, 2016; Seo et al., 2015; Martín-Español et al., 2016; Kopp et al., 2017). Hence, the possibility that the 0.5 m rise in sea level used in scenario SF4 is reached long before the end of the century should be considered. Finally, the uncertainty in both the probability and the consequence are large. Further studies and data collection should therefore be conducted to reduce these uncertainties. Examples include considering dyke breaching and simulating
the combined effect of river floods and storm surges (Zhang et al., 2020).

Differences in the temporal scales of both risks have an impact on the time horizon of risk assessment and consequently on the selection of possible actions to be taken to reduce risk. Results highlight the differences between the hazard consequences of the two risks for the risk owner, with different extent and impact level depending on the hazard severity. The fresh water scarcity can have economic and even social consequences at other risk management levels, as farmers, agro-
industry and local communities, particularly if production is severely affected in quality and quality having impact in related trade and services. Besides the economic impact for the risk owner, inundation can have consequences for farmers if the agricultural land loss is high. Considering the context of the study area, a broader impact of the consequences in agro-industry and local communities can be considered negligible.

The risk assessment approach application in the study area raised some challenges. The definition of both consequence and
risk criteria have to be based on in situ knowledge and historical information. Even if the risk owner has most of the information required, other relevant data are often dispersed in different institutions requiring their aggregation and a prior informed-analysis. The definition of hazard scenarios is another important point to be considered when this approach is applied. As stated before, hazard scenarios have to be anchored on past events information. Valuable information about historical events can be found in a variety of sources, including databases where systematized data are suitable for
supporting risk assessment (Santos, et al., 2014). Several global and national disaster databases are available (e.g. EM-DAT, 2013; DISASTER database, Zêzere et al., 2014) but their resolution is inappropriate for local scale analyses. Regional and local databases are scarcer (e.g. Rilo et al., 2017) but should be used and their development encouraged. The choice of events for the scenarios definition should cover a wide range of consequences and probability, to provide a suitable risk spectrum. Whenever possible scenarios construction should consider the main controlling factors of the hazard severity (e.g.
river discharge, maximum water level, and sea level rise). Monitoring information is crucial in supporting risk management. Timely information will allow the updating of consequence and risk criteria, and hazard scenarios, and will support mitigation and adaptation strategies definition.





As main conclusions, this study presents a risk assessment approach that can be replicated in other agricultural estuarine areas. The approach incorporates scientific-based knowledge of the hazard processes and is suitable to support decision-making at a local level. The consequence descriptors considered can be adapted according to local specificities and updated in time to reflect the evolution of hazard, exposure and vulnerability conditions. At first sight, the extent of the information required to the approach application can be pointed out as a limiting factor. However, the complexity level in both consequence evaluation and criteria definition can be adapted to the available information and tools. Complex numerical models can be used, as in the application to Lezíria Grande presented herein, giving greater scientific robustness to the results. In the absence of this possibility, consequence evaluation and criteria definition can rely on expert judgment supported by past events information. Finally, the risk assessment approach showed to be appropriate to support the discussion of potential mitigation and adaptation measures for risk level reduction, mainly when the possible impact of climate change in risk levels is considered. As future work, the approach is foreseen to be applied to other estuarine agricultural areas and the possible incorporation of further discussion from stakeholders.

**Code and data availability.** The model SCHISM is publicly available at https://github.com/schism-dev/schism.git. The model input files and data are not provided due to the confidentiality of the data.

**Author contribution.** Conceptualization of the risk assessment approach: PF, MR and ABF, with contributions from AF. AF obtained the local data and other information from the risk owner. Application to study area: MR implemented the numerical model, performed the simulations and treated the results for water salinity; ABF implemented the numerical model and performed the simulations for water levels, and PF treated the results for risk assessment. Discussion: all authors contributed. Manuscript preparation: PF prepared the manuscript with contributions from all authors.

**Competing interests.** The authors declare that they have no conflict of interest.

**Acknowledgements.** This work was partially conducted in the scope of the BINGO European H2020 project. The authors thank The Associação de Beneficiários da Lezíria Grande de Vila Franca de Xira (ABLGVFX) for providing data and contributing for discussions about the results.

**Financial support.** This work was partially funded by the BINGO European H2020 project (grant no. 641739).

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

APPENDIX A:

The shallow water model SCHISM (Zhang et al., 2016), including its wave module WWM (Roland et al. 2012), was implemented, calibrated and validated in the Tagus estuary in 2D depth-averaged mode to simulate inundation. At the ocean boundary, the model was forced by tides, surges, and waves taken from regional models. River flows at the upstream boundaries (Tagus and Sorraia) were estimated from data. At the surface, the model is forced by atmospheric pressure and

winds originating from reanalyses. The grid resolution varies between 2 and 800 m. Extensive comparisons with field data showed the model's excellent accuracy, with elevation errors on the order of 10 cm (Fortunato et al., 2017). This accuracy was considered adequate to analyse inundation of the margins under extreme events. To simulate salinity intrusion the system of models SCHISM was also used, but was implemented in 3D baroclinic mode. The numerical model is forced by tides at the oceanic boundary, river flows at the riverine boundaries (Tagus and Sorraia) and atmospheric data at the surface.

The model was previously calibrated and extensively validated in the Tagus estuary against field data (Rodrigues and Fortunato, 2017; Rodrigues et al., 2019). Results showed its ability to represent the circulation and salinity patterns. At Vila Franca de Xira (the station located farther upstream and nearest to Conchoso), in particular, salinity errors were about 2 psu (Rodrigues and Fortunato, 2017). At Conchoso, the Root Mean Square Error and the Mean Absolute Error were 0.4 psu and 0.3 psu, respectively; the model tends to overestimate the data. The maximum difference between the data and the model

results at the peak salinity was about 2 psu (Rodrigues et al., 2019). A detailed description of the model implementation and validation can be found in Rodrigues and Fortunato (2017) and Rodrigues et al. (2019).