# Peer review of "Flood and drought risk assessment for agricultural areas (Tagus Estuary, Portugal)"

_Natural Hazards and Earth System Sciences, 2020_

## Author Comment (AC3)

Reply to reviewer – NHESS-422-2020

**Flood and drought risk assessment for agricultural areas (Tagus Estuary, Portugal)**

The reviewer's comments are in italic. Changes from the original manuscript are marked in blue.

*RC1: This article is highly relevant since it handles a significant and well defined challenge in two dimensions (sea level and river flow) for a very important food production site in Portugal. The scientific contribution is the application of a simple, but consistent and complete risk method that is applicable for the managers of the water supply and irrigation system if the site, including the dikes.*

*I miss a more thorough discussion on how the proposed risk analysed can be used for forecasting analysis and decision support (what type of decision measures they have), when thy should take decisions and how the decisions should be implemented*

Following the reviewer's comment, further details about how the proposed risk assessment tool can support decision and a new table (Table 7) were added to Section 6 Discussion and conclusions (line 465):
""

As directed to support decision-making, the risk assessment approach presented in this paper should be applied together with a risk treatment plan (ISO, 2009). The plan will identify appropriate measures to be taken, in particular to reduce risk when the level of risk is not acceptable or close to. For each specific site, this plan is built upon the knowledge acquired and supported by monitoring and early warning systems. Risk control measures should be identified, evaluated and their acceptance by stakeholders assessed before being applied (Simonovic, 2012). Examples of control measures to cope with water salinity and high water level risks are presented in Table 7. The responsibility for the decision-making and measures implementation will depend on the risk level. Some measures can be implemented by the risk owner and local stakeholders (e.g. farmers); others may depend on the involvement of decision makers and authorities at the national level (e.g. water, agricultural, environment and civil protection authorities). The risk level will determine when each measure should be implemented. An adaptive strategic approach (Mearns, 2010) will be adopted to better deal with uncertainty in the decision making process. Periodic monitoring and review of the risk assessment and treatment processes including the communication and consultation to all involved parts will held. This will contribute to reduce the uncertainty of the process by updating of the risk criteria and risk control measures. The improvement of the knowledge about the system, based on more data and better predictive tools, may also contribute to better characterize, quantify and reduce the uncertainty over time.

Mearns, L.O. (2010). The drama of uncertainty. Climatic Change 100:77–85. DOI 10.1007/s10584-010-9841-6

Table 7. Examples of risk control measures concerning water salinity and high water level risks.

| Risk | Measure | Responsible for decision making / implementation | When the implementation should take place |
|------|---------|--------------------------------------------------|-------------------------------------------|
| **Water salinity** | Extract fresh water from alternative source | Risk owner / Risk owner and local stakeholders | When the level of risk is tolerable but tends to increase |
| | Reuse irrigation water | Risk owner / Risk owner and local stakeholders | When the level of risk is tolerable but tends to increase |
| | Adapt crops (higher salt tolerance, less water demanding, shorter growth period) | Risk owner / Risk owner and local stakeholders | When the level of risk is intolerable |
| | Construct reservoir | Risk owner and National authorities / Risk owner and National authorities | When the level of risk is intolerable |
| **High water level** | Implement flood monitoring and early warning systems | Risk owner and National authorities / Risk owner and National authorities | Immediately, to support risk management |
| | Raise dyke level | Risk owner / Risk owner | When the level of risk is tolerable but tends to increase |
| | Reinforce dyke | Risk owner / Risk owner and Environment and Agricultural authorities | When the level of risk is tolerable but tends to increase |
| | Transfer valuable goods and infrastructures to other areas | Risk owner / Risk owner | When the level of risk is tolerable but tends to increase |
| | Implement a water retention basin along the dyke | Risk owner and Environment and Agricultural authorities / Risk owner and Environment and Agricultural authorities | When the level of risk is intolerable |
| | Create new artificial wetlands | Risk owner and Environment and Agricultural authorities / Risk owner and Environment and Agricultural authorities | When the level of risk is intolerable |

---

## Author Comment (AC4)

Reply to reviewer – NHESS-422-2020

**Flood and drought risk assessment for agricultural areas (Tagus Estuary, Portugal)**

The reviewer's comments are in italic. Changes from the original manuscript are marked in blue.

*RC2: Please make a broader and more detailed explanation to figure 3 and 4. Please check figure 4 itself versus figure text (not in accordance). Please explain the lines drawn within various subgraphs of figure 3.*

Following the reviewer's comment, further details were added to the discussion about figures 3 and 4 as follows:
"Figure 3 presents the different scenarios projected in the consequence/probability diagram for the water unavailability; the horizontal and vertical bars represent the expected uncertainty for consequence and likelihood, respectively. The uncertainty of the consequence was estimated considering that the model overestimates the measured salinity by up to 2 psu. Hence, for each scenario the uncertainty was calculated assuming the maximum tolerable salinity in the water for irrigation as 3 psu (i.e., the maximum tolerable salinity, taken as 1 psu, plus the maximum error).
Consequence is low for all the scenarios in the first week, since the water available fulfils all the needs for irrigation. As time progresses (and the river flow remains constant) the consequence increases for all the scenarios with exception of scenario SD1 (climatological, mean river flow of 132 $m^3.s^{-1}$), in which water is always available for irrigation. For scenario SD2 (river flow of 44 $m^3.s^{-1}$) the consequence is moderate in week 3 and about 90% of the water needed for irrigation is available. In week 4 the water available for irrigation decreases to about 20% of the needs in this scenario (Fig. 3). The consequences are also more severe when the river flow is lower, as expected. For scenarios SD3 (river flow of 22 $m^3.s^{-1}$), SD4 (river flow of 16.5 $m^3.s^{-1}$) and SD5 (river flow of 8 $m^3.s^{-1}$) freshwater is unavailable for irrigation in week 3 (Fig. 3). However, the very low river flow scenarios (SD4, SD5) have low likelihoods. The estimated consequences for the scenarios agree with the observed occurrences during recent droughts (2005, 2012), as described by the risk owner. During July and August of both 2012 and 2005, droughts represented by scenarios SD2 and SD3 respectively, salinity reached concentrations at the Conchoso water intake that were inadequate for irrigation. In 2012, in particular, water with salinity of about 1.1–1.2 was used for irrigation, which reduced the production. However, the adverse impacts of the 2005 drought were more severe for the farmers in the Lezíria, since the drought itself was more severe and the ABLGVFX had fewer resources and was less prepared to deal with these events. More severe consequences are also estimated for scenario SD3 comparatively to scenario SD2 (Fig. 3). The comparison between scenarios SD3 (river flow of 22 $m^3.s^{-1}$) and SD6 (river flow of 22 $m^3.s^{-1}$ and mean SLR of 0.5 m) suggests that, for the same river flow, SLR increases the consequences (Fig.3).
Since the consequence of all the scenarios is estimated based on numerical simulations there is an associated uncertainty.  Results suggest that the uncertainty associated with the numerical simulations on the consequence severity is higher for low river flow scenarios. In some cases, consequences can range from

"Very high" to "Low". However, this larger variability is explained by the criterion used to define the uncertainty (the maximum peak difference).

Regarding the risk diagram, results indicate that for all the scenarios except for the climatological scenario (SD1) the risk is intolerable in the last week (Fig. 4). Risk also grows with the duration of the droughts: for instance, for scenarios SD2 (river flow of 44 $m^3.s^{-1}$; return period of 5-10 years) and SD3 (river flow of 22 $m^3.s^{-1}$; return period of 10-100 years) risk can be medium until the third and second weeks respectively, and intolerable if the drought lasts for longer periods (Fig. 4). In these cases, when the river flow remains low for several consecutive weeks, even using the Risco River as an alternative freshwater source is insufficient to meet the irrigation needs. For the remaining river flow alone scenarios (scenarios SD4 and SD5) the risk is intolerable as early as the second week (Fig. 4); however the return period of these events is estimated to be larger than 100 years and their likelihood is, consequently, low. For events similar to scenarios SD2 and SD3, risk treatment is mandatory to reduce the risk level and may include the use of alternative water sources, the selection of alternative crops, the reduction of the irrigated area and/or the construction of water storage facilities. Mean SLR may represent an additional source of risk (scenario SD6, Fig. 4) and should also be taken into account in the establishment of risk management and climate change adaption plans for this agricultural area.

Figure 4 was corrected because the color scheme for the weeks was not in accordance with the figure's caption and the captions of figures 3 and 4 were also changed as follows:

Figure 3. Consequence/probability diagrams for water unavailability for irrigation during weeks 1 to 4. The river flow is constant during all weeks. The river flows considered in each scenario are: SD1 – 132 $m^3.s^{-1}$; SD2 – 44 $m^3.s^{-1}$; SD3 – 22 $m^3.s^{-1}$; SD4 – 16.5 $m^3.s^{-1}$; SD5 – 8 $m^3.s^{-1}$; SD6 – 22 $m^3.s^{-1}$ and mean sea level rise of 0.5 m. Error bars represent the uncertainty in the likelihood and in the consequence.

[Figure]

Figure 4. Risk for water unavailability. Colours of the symbols represent the weeks (darker to lighter means week 1 to week 4). The river flows considered in each scenario are: SD1 – 132 $m^3.s^{-1}$; SD2 – 44 $m^3.s^{-1}$; SD3 – 22 $m^3.s^{-1}$; SD4 – 16.5 $m^3.s^{-1}$; SD5 – 8 $m^3.s^{-1}$; SD6 – 22 $m^3.s^{-1}$ and mean sea level rise of 0.5 m. The following events are not represented in the risk diagram because all the water needed for irrigation is available and the consequence is 0: scenario SD1 – all weeks; scenario SD2 – weeks 1 and 2; scenarios SD3, SD4, SD5 and SD6 – week 1.

---

## Author Comment (AC5)

Reply to reviewer – NHESS-422-2020

**Flood and drought risk assessment for agricultural areas (Tagus Estuary, Portugal)**

The reviewer's comments are in italic. Changes from the original manuscript are marked in blue.

*RC3: This is an interesting and relevant study, and the authors have selected a suitable method for flood and drought risk assessment for agricultural areas. The consequence/probability diagram is a suitable method for this study and is well presented; however, the study would be stronger if it included more than a risk assessment. The paper is generally well written and structured. Though, the paper has some shortcomings regarding the treatment and monitoring of risk, which should be included.*

*I miss a more thorough discussion on how the approach can be used in risk management, as the method covers risk assessment but is missing risk treatment and the process of monitoring and modify risk in accordance with ISI 31000. The paper would be stronger if more efforts were added to include risk management (risk treatment, monitoring, and communication), especially since risk management is given so much space in the introduction and the objectives of the study.*

*Please include further details (perhaps in the discussion) on how the risk can be managed and be used for decision making (To follow up the author's recommendation that the risk owner should consider risk reduction measures in line 438). (Or remove/rewrite line 55-56 describing the tool to support the management of risk at a local level)*

*Is miss a discussion of the uncertainty, as briefly discussed in line 424, as this is one of the two main challenges presented in the introduction (line 55). An a more detailed discussion of uncertainty and uncertainty reduction would strengthen the paper.*

Following the reviewer's comments regarding the treatment and monitoring of risk, further details were added to Section 6 Discussion and conclusions (line 465). A new table (Table 7) presenting examples of risk control measures was also added. In what concerns the uncertainty, it was now stressed in the discussion that over time, the periodic monitoring and review of the risk management process with more data and better predictive tools, is expected to contribute to better characterize, quantify and reduce the uncertainty.

As directed to support decision-making, the risk assessment approach presented in this paper should be applied together with a risk treatment plan (ISO, 2009). The plan will identify appropriate measures to be taken, in particular to reduce risk when the level of risk is not acceptable or close to. For each specific site, this plan is built upon the knowledge acquired and supported by monitoring and early warning systems. Risk control measures should be identified, evaluated and their acceptance by stakeholders assessed before being applied (Simonovic, 2012). Examples of control measures to cope with water salinity and high water level risks are presented in Table 7. The responsibility for the decision-making and measures implementation

will depend on the risk level. Some measures can be implemented by the risk owner and local stakeholders (e.g. farmers); others may depend on the involvement of decision makers and authorities at the national level (e.g. water, agricultural, environment and civil protection authorities). The risk level will determine when each measure should be implemented. An adaptive strategic approach (Mearns, 2010) will be adopted to better deal with uncertainty in the decision making process. Periodic monitoring and review of the risk assessment and treatment processes including the communication and consultation to all involved parts will held. This will contribute to reduce the uncertainty of the process by updating of the risk criteria and risk control measures. The improvement of the knowledge about the system, based on more data and better predictive tools, may also contribute to better characterize, quantify and reduce the uncertainty over time.

Mearns, L.O. (2010). The drama of uncertainty. Climatic Change 100:77–85. DOI 10.1007/s10584-010-9841-6

Table 7. Examples of risk control measures concerning water salinity and high water level risks.

| Risk | Measure | Responsible for decision making / implementation | When the implementation should take place |
|------|---------|--------------------------------------------------|-------------------------------------------|
| **Water salinity** | Extract fresh water from alternative source | Risk owner / Risk owner and local stakeholders | When the level of risk is tolerable but tends to increase |
| | Reuse irrigation water | Risk owner / Risk owner and local stakeholders | When the level of risk is tolerable but tends to increase |
| | Adapt crops (higher salt tolerance, less water demanding, shorter growth period) | Risk owner / Risk owner and local stakeholders | When the level of risk is intolerable |
| | Construct reservoir | Risk owner and National authorities / Risk owner and National authorities | When the level of risk is intolerable |
| **High water level** | Implement flood monitoring and early warning systems | Risk owner and National authorities / Risk owner and National authorities | Immediately, to support risk management |
| | Raise dyke level | Risk owner / Risk owner | When the level of risk is tolerable but tends to increase |
| | Reinforce dyke | Risk owner / Risk owner and Environment and Agricultural authorities | When the level of risk is tolerable but tends to increase |
| | Transfer valuable goods and infrastructures to other areas | Risk owner / Risk owner | When the level of risk is tolerable but tends to increase |
| | Implement a water retention basin along the dyke | Risk owner and Environment and Agricultural authorities / Risk owner and Environment and Agricultural authorities | When the level of risk is intolerable |
| | Create new artificial wetlands | Risk owner and Environment and Agricultural authorities / Risk owner and Environment and Agricultural authorities | When the level of risk is intolerable |

*Please improve captions of Figure 3 and 4. Figure captions should be standalone, not dependent on explanation in the text. For Figure 3 Week 1 you could consider different scale to improve readability.*

Following the reviewer's comment, more information in the captions of Figure 3 and 4 was included as follows:

[revised manuscript text omitted]

---

## Author Response (AR1)

**Author's response - NHESS-422-2020: Flood and drought risk assessment for agricultural areas (Tagus Estuary, Portugal)**

We thank the editor and reviewers for their comments. We have incorporated changes to reflect the reviewer's suggestions and a point-by-point reply is presented below. The reviewers' comments are in italic and changes from the original manuscript are marked in blue.

*RC1 reviewer: This article is highly relevant since it handles a significant and well defined challenge in two dimensions (sea level and river flow) for a very important food production site in Portugal. The scientific contribution is the application of a simple, but consistent and complete risk method that is applicable for the managers of the water supply and irrigation system if the site, including the dikes.*

*I miss a more thorough discussion on how the proposed risk analysed can be used for forecasting analysis and decision support (what type of decision measures they have), when thy should take decisions and how the decisions should be implemented*

Following the **RC1** reviewer's comment, further details about how the proposed risk assessment tool can support decision and a new table (Table 7) were added to Section 6 Discussion and conclusions (line 465).

 As directed to support decision-making, the risk assessment approach presented here should be applied together with a risk treatment plan (ISO, 2009). The plan will identify appropriate measures to be taken, in particular to reduce risk when the level of risk approaches or exceeds an unacceptability threshold. For each specific site, this plan is built upon the knowledge acquired and supported by monitoring and early warning systems. Risk control measures should be identified, evaluated and accepted by stakeholders before being applied (Simonovic, 2012). Examples of control measures to cope with water salinity and high water level risks are presented in Table 7. The responsibility for the decision-making and measures implementation will depend on the risk level. Some measures can be implemented by the risk owner and local stakeholders (e.g. farmers); others may require the involvement of decision-makers and authorities at the national level (e.g. water, agricultural, environment and civil protection authorities). The risk level determines when each measure should be implemented. An adaptive strategic approach (Mearns, 2010) will be adopted to better deal with uncertainty in the decision-making process. Periodic monitoring and review of the risk assessment and treatment processes, including the communication and consultation to all involved parts, will held. This approach will contribute to reduce the uncertainty of the process by updating the risk criteria and risk control measures. The improvement of the knowledge about the system, based on more data and better predictive tools, may also contribute to better characterize, quantify and reduce the uncertainty over time.

A new reference was added to the reference list:

Mearns, L.O. (2010). The drama of uncertainty. Climatic Change 100:77–85. DOI 10.1007/s10584-010-9841-6

Table 7. Examples of risk control measures concerning water salinity and high water level risks.

| Risk | Measure | Responsible for decision making / implementation | When the implementation should take place |
|---|---|---|---|
| **Water salinity** | Extract fresh water from an alternative source | Risk owner / Risk owner and local stakeholders | When the level of risk is tolerable but rising |
| | Reuse irrigation water | Risk owner / Risk owner and local stakeholders | When the level of risk is tolerable but rising |
| | Adapt crops (higher salt tolerance, less water demanding, shorter growth period) | Risk owner / Risk owner and local stakeholders | When the level of risk is intolerable |
| | Construct reservoir | Risk owner and National authorities / Risk owner and National authorities | When the level of risk is intolerable |
| **High water level** | Implement flood monitoring and early warning systems | Risk owner and National authorities / Risk owner and National authorities | Immediately, to support risk management |
| | Raise dyke level | Risk owner / Risk owner | When the level of risk is tolerable but rising |
| | Reinforce dyke | Risk owner / Risk owner and Environment and Agricultural authorities | When the level of risk is tolerable but rising |
| | Transfer valuable goods and infrastructures to other areas | Risk owner / Risk owner | When the level of risk is tolerable but rising |
| | Implement a water retention basin along the dyke | Risk owner and Environment and Agricultural authorities / Risk owner and Environment and Agricultural authorities | When the level of risk is intolerable |
| | Create new artificial wetlands | Risk owner and Environment and Agricultural authorities / Risk owner and Environment and Agricultural authorities | When the level of risk is intolerable |

*RC2 reviewer: Please make a broader and more detailed explanation to figure 3 and 4. Please check figure 4 itself versus figure text (not in accordance). Please explain the lines drawn within various subgraphs of figure 3.*

Following the reviewer's comment, further details were added to the discussion about figures 3 and 4 as follows:

[revised manuscript text omitted]

***RC3 reviewer:*** *This is an interesting and relevant study, and the authors have selected a suitable method for flood and drought risk assessment for agricultural areas. The consequence/probability diagram is a suitable method for this study and is well presented; however, the study would be stronger if it included more than a risk assessment. The paper is generally well written and structured. Though, the paper has some shortcomings regarding the treatment and monitoring of risk, which should be included.*

*I miss a more thorough discussion on how the approach can be used in risk management, as the method covers risk assessment but is missing risk treatment and the process of monitoring and modify risk in accordance with ISI 31000. The paper would be stronger if more efforts were added to include risk management (risk treatment, monitoring, and communication), especially since risk management is given so much space in the introduction and the objectives of the study.*

*Please include further details (perhaps in the discussion) on how the risk can be managed and be used for decision making (To follow up the author's recommendation that the risk owner should consider risk reduction measures in line 438). (Or remove/rewrite line 55-56 describing the tool to support the management of risk at a local level)*

Following the reviewer's comments regarding the treatment and monitoring of risk, further details were added to Section 6 Discussion and conclusions (line 465) and new table (Table 7) presenting examples of risk control measures was also added. Please, see our response to RC1 reviewer.

*Is miss a discussion of the uncertainty, as briefly discussed in line 424, as this is one of the two main challenges presented in the introduction (line 55). An a more detailed discussion of uncertainty and uncertainty reduction would strengthen the paper.*

Regarding uncertainty, the discussion now stresses that over time, the periodic monitoring and review of the risk management process with more data and better predictive tools, is expected to contribute to better characterize, quantify and reduce the uncertainty.

*Please improve captions of Figure 3 and 4. Figure captions should be standalone, not dependent on explanation in the text. For Figure 3 Week 1 you could consider different scale to improve readability.*

Following the reviewer's comment, more information in the captions of Figure 3 and 4 was included. Please, see our response to RC2 reviewer.